# Hyperbolic Self-paced Learning for Self-supervised Skeleton-based Action Representations

**Luca Franco**[†1]  **Paolo Mandica**[†1]  **Bharti Munjal**[1,2]  **Fabio Galasso**[1]
[1]Sapienza University of Rome  [2]Technical University of Munich

## Abstract

Self-paced learning has been beneficial for tasks where some initial knowledge is available, such as weakly supervised learning and domain adaptation, to select and order the training sample sequence, from easy to complex. However its applicability remains unexplored in unsupervised learning, whereby the knowledge of the task matures during training.

We propose a novel HYperbolic Self-Paced model (HYSP) for learning skeleton-based action representations. HYSP adopts self-supervision: it uses data augmentations to generate two views of the same sample, and it learns by matching one (named *online*) to the other (the *target*). We propose to use hyperbolic uncertainty to determine the algorithmic learning pace, under the assumption that less uncertain samples should be more strongly driving the training, with a larger weight and pace. Hyperbolic uncertainty is a by-product of the adopted hyperbolic neural networks, it matures during training and it comes with no extra cost, compared to the established Euclidean SSL framework counterparts.

When tested on three established skeleton-based action recognition datasets, HYSP outperforms the state-of-the-art on PKU-MMD I, as well as on 2 out of 3 downstream tasks on NTU-60 and NTU-120. Additionally, HYSP only uses positive pairs and bypasses therefore the complex and computationally-demanding mining procedures required for the negatives in contrastive techniques.
Code is available at `https://github.com/paolomandica/HYSP`.

## 1 Introduction

Starting from the seminal work of Kumar et al. (2010), the machine learning community has started looking at self-paced learning, i.e. determining the ideal sample order, from easy to complex, to improve the model performance. Self-paced learning has been adopted so far for weakly-supervised learning (Liu et al., 2021; Wang et al., 2021; Sangineto et al., 2019), or where some initial knowledge is available, e.g. from a source model, in unsupervised domain adaption (Liu et al., 2021). Self-paced approaches use the label (or pseudo-label) confidence to select easier samples and train on those first. However labels are not available in self-supervised learning (SSL) (Chen et al., 2020a; He et al., 2020; Grill et al., 2020; Chen & He, 2021), where the supervision comes from the data structure itself, i.e. from the sample embeddings.

We propose HYSP, the first HYperbolic Self-Paced learning model for SSL. In HYSP, the self-pacing confidence is provided by the hyperbolic uncertainty (Ganea et al., 2018; Shimizu et al., 2021) of each data sample. In more details, we adopt the Poincaré Ball model (Surís et al., 2021; Ganea et al., 2018; Khrulkov et al., 2020; Ermolov et al., 2022) and define the certainty of each sample as its embedding radius. The hyperbolic uncertainty is a property of each data sample in hyperbolic space, and it is therefore available while training with SSL algorithms.

HYSP stems from the belief that the uncertainty of samples matures during the SSL training and that more certain ones should drive the training more prominently, with a larger pace, at each stage of training. In fact, hyperbolic uncertainty is trained end-to-end and it matures as the training proceeds,

---

[†]Equal contribution

i.e. data samples become more certain. We consider the task of human action recognition, which has drawn growing attention (Singh et al., 2021; Li et al., 2021; Guo et al., 2022a; Chen et al., 2021a; Kim et al., 2021) due to its vast range of applications, including surveillance, behavior analysis, assisted living and human-computer interaction, while being skeletons convenient light-weight representations, privacy preserving and generalizable beyond people appearance (Xu et al., 2020; Lin et al., 2020a).

HYSP builds on top of a recent self-supervised approach, SkeletonCLR (Li et al., 2021), for training skeleton-based action representations. HYSP generates two views for the input samples by data augmentations (He et al., 2020; Chen et al., 2020a; Caron et al., 2020; Grill et al., 2020; Chen & He, 2021; Li et al., 2021), which are then processed with two Siamese networks, to produce two sample representations: an *online* and a *target*. The training proceeds by tasking the former to match the latter, being both of them *positives*, i.e. two *views* of the same sample. HYSP only considers positives during training and it requires curriculum learning. The latter stems from the vanishing gradients of hyperbolic embeddings upon initialization, due to their initial overconfidence (high radius) (Guo et al., 2022b). So initially, we only consider angles, which coincides with starting from the conformal Euclidean optimization. The former is because matching two embeddings in hyperbolic implies matching their uncertainty too, which appears ill-posed for negatives from different samples, i.e. uncertainty is specific of each sample at each stage of training. This bypasses the complex and computationally-demanding procedures of negative mining, which contrastive techniques require[*]. Both aspects are discussed in detail in Sec. 3.2.

We evaluate HYSP on three most recent and widely-adopted action recognition datasets, NTU-60, NTU-120 and PKU-MMD I. Following standard protocols, we pre-train with SSL, then transfer to a downstream skeleton-base action classification task. HYSP outperforms the state-of-the-art on PKU-MMD I, as well as on 2 out of 3 downstream tasks on NTU-60 and NTU-120.

## 2 RELATED WORK

HYSP embraces work from four research fields, for the first time, as we detail in this section: self-paced learning, hyperbolic neural networks, self-supervision and skeleton-based action recognition.

### 2.1 SELF-PACED LEARNING

Self-paced learning (SPL), initially introduced by Kumar et al. (2010) is an extension of curriculum learning (Bengio et al., 2009) which automatically orders examples during training based on their difficulty. Methods for self-paced learning can be roughly divided into two categories. One set of methods (Jiang et al., 2014; Wu et al., 2021) employ it for fully supervised problems. For example, Jiang et al. (2014); Wu et al. (2021) employ it for image classification. Jiang et al. (2014) enhance SPL by considering the diversity of the training examples together with their hardness to select samples. Wu et al. (2021) studies SPL when training with limited time budget and noisy data. The second category of methods employ it for weakly or semi supervised learning. Methods in Sanghineto et al. (2019); Zhang et al. (2016a) adopt it for weakly supervised object detection, where Sanghineto et al. (2019) iteratively selects the most reliable images and bounding boxes, while Zhang et al. (2016a) uses saliency detection for self-pacing. Methods in Peng et al. (2021); Wang et al. (2021) employ SPL for semi-supervised segmentation. Peng et al. (2021) adds a regularization term in the loss to learn importance weights jointly with network parameters. Wang et al. (2021) considers the prediction uncertainty and uses a generalized Jensen Shannon Divergence loss. Both categories require the notion of classes and not apply to SSL frameworks, where sample embeddings are only available.

### 2.2 HYPERBOLIC NEURAL NETWORKS

Hyperbolic representation learning in deep neural networks gained momentum after the pioneering work hyperNNs (Ganea et al., 2018), which proposes hyperbolic counterparts for the classical (Euclidean) fully-connected layers, multinomial logistic regression and RNNs. Other representative work has introduced hyperbolic convolution neural networks (Shimizu et al., 2021), hyperbolic

---

[*]Similarly to BYOL (Grill et al., 2020), HYSP is not a contrastive technique, as it only adopts positives.

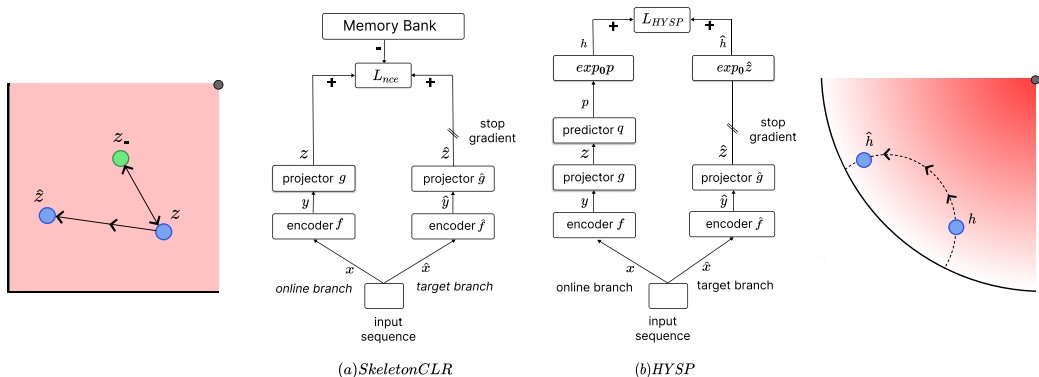

Figure 1: (a) SkeletonCLR is a contrastive learning approach based on pushing the embeddings of two views $x$ and $\hat{x}$ of the same sample (*positives*) close to each other, while repulsing from embeddings of other samples (*negatives*). In the represented Euclidean space, on the left side, the attracting/repulsive force is a straight line. (b) The proposed HYSP maps the embeddings into a hyperbolic space, where the sample uncertainty determines the learning pace. The hyperbolic Poincaré ball, on the right side, illustrates the embeddings, where the radius indicates the uncertainty. The attractive force proceeds on a circle, orthogonal to the boundary circle. So the force grows polynomially with the radius. In HYSP, learning is paced by the uncertainty, which the model learns end-to-end during SSL.

graph neural networks (Liu et al., 2019; Chami et al., 2019), and hyperbolic attention networks (Gulcehre et al., 2019). Two distinct aspects have motivated hyperbolic geometry: their capability to encode hierarchies and tree-like structures Chami et al. (2020); Khrulkov et al. (2020), also inspiring Peng et al. (2020) for skeleton-based action recognition; and their notion of uncertainty, encoded by the embedding radius Surís et al. (2021); Atigh et al. (2022), which we leverage here. More recently, hyperbolic geometry has been used with self-supervision (Yan et al., 2021; Surís et al., 2021; Ermolov et al., 2022). Particularly, Surís et al. (2021) and Ermolov et al. (2022) let hierarchical actions and image categories emerge from the unlabelled data using RNNs and Vision Transformers, respectively. Differently, our HYSP uses GCNs and it applies to skeleton data. Moreover, unlike all prior works, HYSP leverages hyperbolic uncertainty to drive the training by letting more certain pairs steer the training more.

## 2.3 SELF-SUPERVISED LEARNING

Self-supervised learning aims to learn discriminative feature representations from unlabeled data. The supervisory signal is typically derived from the input using certain *pretext* tasks and it is used to pre-train the encoder, which is then transferred to the downstream task of interest. Among all the proposed methods (Noroozi & Favaro, 2016; Zhang et al., 2016b; Gidaris et al., 2018; Caron et al., 2018; 2020), a particularly effective one is the contrastive learning, introduced by SimCLR Chen et al. (2020a). Contrastive learning brings different views of the same image closer ('positive pairs'), and pushes apart a large number of different ones ('negative pairs'). MoCo (He et al., 2020; Chen et al., 2020b) partially alleviates the requirement of a large batch size by using a memory-bank of negatives. BYOL (Grill et al., 2020) pioneers on training with positive pairs only. Without the use of negatives, the risk of collapsing to trivial solutions is avoided by means of a predictor in one of the Siamese networks, and a momentum-based encoder with stop-gradient in the other. SimSiam (Chen & He, 2021) further simplifies the design by removing the momentum encoder. Here we draw on the BYOL design and adopt positive-only pairs for skeleton-based action recognition. We experimentally show that including negatives harm training self-supervised hyperbolic neural networks.

## 2.4 SKELETON-BASED ACTION RECOGNITION

Most recent work in (supervised) skeleton-based action recognition (Chen et al., 2021a;b; Wang et al., 2020) adopts the ST-GCN model of Yan et al. (2018) for its simplicity and performance. We

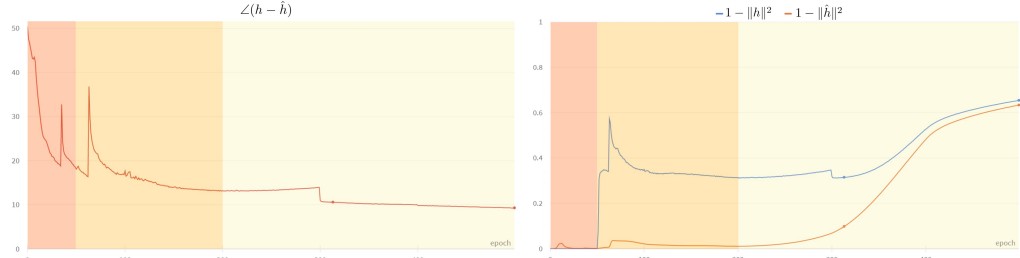

Figure 2: Analysis of the parts of eq. 5, namely (*left*) the angle between $h$ and $\hat{h}$ and (*right*) the quantity $(1-\|h\|^2)$, monotonic with the embedding uncertainty $(1-\|h\|)$, during the training phases, i.e. initial (orange), intermediate (yellow), and final (light yellow). See discussion in Sec. 3.2.1.

also use ST-GCN. The self-supervised skeleton-based action recognition methods can be broadly divided into two categories. The first category uses encoder-decoder architectures to reconstruct the input (Zheng et al., 2018; Su et al., 2020; Paoletti et al., 2021; Yang et al., 2021; Su et al., 2021). The most representative example is AE-L (Paoletti et al., 2021), which uses Laplacian regularization to enforce representations aware of the skeletal geometry. The other category of methods (Lin et al., 2020a; Thoker et al., 2021; Li et al., 2021; Rao et al., 2021) uses contrastive learning with negative samples. The most representative method is SkeletonCLR (Li et al., 2021) which uses momentum encoder with memory-augmented contrastive learning, similar to MoCo. More recent techniques, CrosSCLR (Li et al., 2021) and AimCLR (Guo et al., 2022a), are based on SkeletonCLR. The former extends it by leveraging multiple views of the skeleton to mine more positives; the latter by introducing extreme augmentations, features dropout, and nearest neighbors mining for extra positives. We also build on SkeletonCLR, but we turn it into positive-only (BYOL), because only uncertainty between positives is justifiable (cf. Sec. FGX-fill-in). BYOL for skeleton-based SSL has first been introduced by Moliner et al. (2022). Differently from them, HYSP introduces self-paced SSL learning with hyperbolic uncertainty.

## 3 METHOD

We start by motivating hyperbolic self-paced learning, then we describe the baseline SkeletonCLR algorithm and the proposed HYSP. We design HYSP according to the principle that more certain samples should drive the learning process more predominantly. Self-pacing is determined by the sample target embedding, which the sample online embeddings are trained to match. Provided the same distance between target and online values, HYSP gradients are larger for more certain targets[†]. Since in SSL no prior information is given about the samples, uncertainty needs to be estimated end-to-end, alongside the main objective. This is realized in HYSP by means of a hyperbolic mapping and distance, combined with the positives-only design of BYOL (Grill et al., 2020) and a curriculum learning (Bengio et al., 2009) which gradually transitions from the Euclidean to the uncertainty-aware hyperbolic space (Surís et al., 2021; Khrulkov et al., 2020).

### 3.1 BACKGROUND

A few most recent techniques (Guo et al., 2022a; Li et al., 2021) for self-supervised skeleton representation learning adopt SkeletonCLR Li et al. (2021). That is a contrastive learning approach, which forces two augmented views of an action instance (positives) to be embedded closely, while repulsing different instance embeddings (negatives).

Let us consider Fig. 1(a). The positives $x$ and $\hat{x}$ are two different augmentations of the same action sequence, e.g. subject to shear and temporal crop. The samples are encoded into $y = f(x, \theta)$ and $\hat{y} = \hat{f}(\hat{x}, \hat{\theta})$ by two Siamese networks, an online $f$ and a target $\hat{f}$ encoder with parameters $\theta$ and $\hat{\theta}$ respectively. Finally the embeddings $z$ and $\hat{z}$ are obtained by two Siamese projector MLPs, $g$ and $\hat{g}$.

---

[†]Gradients generally depend on the distance between the estimation (here online) and the reference (here target) ground truth, as a function of the varying estimation. In self-paced learning, references vary during training.

SkeletonCLR also uses negative samples which are embeddings of non-positive instances, queued in a memory bank after each iteration, and dequeued according to a first-in-first-out principle. Positives and negatives are employed within the Noise Contrastive Estimation loss (Oord et al., 2018):

$$L_{nce} = -\log \frac{\exp(z \cdot \hat{z}/\tau)}{\exp(z \cdot \hat{z}/\tau) + \sum_{i=1}^{M} \exp(z \cdot m_i/\tau)} \tag{1}$$

where $z \cdot \hat{z}$ is the normalised dot product (cosine similarity), $m_i$ is a negative sample from the queue and $\tau$ is the softmax temperature hyper-parameter.

Two aspects are peculiar of SkeletonCLR, important for its robustness and performance: the momentum encoding for $\hat{f}$ and the stop-gradient on the corresponding branch. In fact, the parameters of the target encoder $\hat{f}$ are not updated by back-propagation but by an exponential moving average from the online encoder, i.e $\hat{\theta} \leftarrow \alpha\hat{\theta} + (1-\alpha)\theta$, where $\alpha$ is the momentum coefficient. A similar notation and procedure is adopted for $\hat{g}$. Finally, stop-gradient enables the model to only update the online branch, keeping the other as the reference "target".

### 3.2 HYPERBOLIC SELF-PACED LEARNING

Basing on SkeletonCLR, we propose to learn by comparing the online $h$ and target $\hat{h}$ representations in hyperbolic embedding space. We do so by mapping the outputs of both branches into the Poincaré ball model centered at $\mathbf{0}$ by an *exponential map* (Ganea et al., 2018):

$$\hat{h} = Exp_0^c(\hat{z}) = tanh(\sqrt{c}\,\|\hat{z}\|)\frac{\hat{z}}{\sqrt{c}\,\|\hat{z}\|} \tag{2}$$

where $c$ represents the curvature of the hyperbolic space and $\|\cdot\|$ the standard Euclidean $L^2$-norm. $h$ and $\hat{h}$ are compared by means of the Poincaré distance:

$$L_{poin}(h,\hat{h}) = \cosh^{-1}\left(1 + 2\frac{\|h-\hat{h}\|^2}{(1-\|h\|^2)(1-\|\hat{h}\|^2)}\right) \tag{3}$$

with $\|h\|$ and $\|\hat{h}\|$ the radii of $h$ and $\hat{h}$ in the Poincaré ball.

Following Surís et al. (2021)[‡], we define the hyperbolic uncertainty of an embedding $\hat{h}$ as:

$$u_{\hat{h}} = 1 - \|\hat{h}\| \tag{4}$$

Note that the hyperbolic space volume increases with the radius, and the Poincaré distance is exponentially larger for embeddings closer to the hyperbolic ball edge, i.e. the matching of *certain* embeddings is penalized exponentially more. Upon training, this yields larger uncertainty for more ambiguous actions, cf. Sec. 4.4 and Appendix A. Next, we detail the self-paced effect on learning.

#### 3.2.1 UNCERTAINTY FOR SELF-PACED LEARNING.

Learning proceeds by minimizing the Poincaré distance via stochastic Riemannian gradient descent (Bonnabel, 2013). This is based on the Riemannian gradient of Eq. 3, computed w.r.t. the online hyperbolic embedding $h$, which the optimization procedure pushes to match the target $\hat{h}$:

$$\nabla L_{poin}(h,\hat{h}) = \frac{(1-\|h\|^2)^2}{2\sqrt{(1-\|h\|^2)(1-\|\hat{h}\|^2)+\|h-\hat{h}\|^2}}\left(\frac{h-\hat{h}}{\|h-\hat{h}\|} + \frac{h\|h-\hat{h}\|}{1-\|h\|^2}\right) \tag{5}$$

Learning is *self-paced*, because the gradient changes according to the certainty of the target $\hat{h}$, i.e. the larger their radius $\|\hat{h}\|$, the more certain $\hat{h}$, and the stronger are the gradients $\nabla L_{poin}(h,\hat{h})$, irrespective of $h$. We further discuss the training dynamics of Eq. 5. Let us consider Fig. 2, plotting for each training epoch the angle difference between $h$ and $\hat{h}$, namely $\angle(h-\hat{h})$, and respectively the quantities $1-\|h\|^2$ and $1-\|\hat{h}\|^2$:

---

[‡]The definition is explicitly adopted in their code. This choice differs from Khrulkov et al. (2020) which uses the Poincaré distance, i.e. $u_{\hat{h}} = 1 - L_{poin}(\hat{h}, 0)$. However both definitions provide similar rankings of uncertainty, which matters here. We regard this difference as a calibration issue, which we do not investigate.

- At the beginning of training (Fig. 2 orange section), following random initialization, the embeddings are at the edge of the Poincaré ball $(1-\|h\|^2, 1-\|\hat{h}\|^2 \approx 0)$. Assuming that the representations of the two views be still not close $(\|h-\hat{h}\| > 0)$, the gradient $\nabla L_{poin}(h, \hat{h})$ vanishes (Guo et al., 2022b). The risk of stall is avoided by means of curriculum learning (see Sec. 3.2.3), which considers only their angle for the first 50 epochs;

- At intermediate training phases (Fig. 2 yellow section), the radius of $h$ is also optimized. $h$ departs from the ball edge $(1 - \|h\|^2 > 0$ in Fig. 2b) and it moves towards $\hat{h}$, marching along the geodesic $(\|h-\hat{h}\| > 0)$. At this time, learning is self-paced, i.e. the importance of each sample is weighted according to the certainty of the target view: the higher the radius of $\hat{h}$, the larger the gradient. Note that the online embedding $h$ is pushed towards the ball edge, since $\hat{h}$ is momentum-updated and it moves more slowly away from the ball edge, where it was initialized. This prevents the embeddings from collapsing to trivial solutions during epochs 50-200;

- The training ends (Fig. 2 light yellow section) when $\hat{h}$ also departs from the ball edge, due to the updates of the model encoder, brought up by $h$ with momentum. Both $h$ and $\hat{h}$ draws to the ball origin, collapsing into the trivial null solution.

Next, we address two aspects of the hyperbolic optimization: **i.** the use of negatives is not well-defined in the hyperbolic space and, in fact, leveraging them deteriorates the model performance; **ii.** the exponential penalization of the Poincaré distance results in gradients which are too small at early stages, causing vanishing gradient. We address these challenges by adopting relatively positive pairs only (Grill et al., 2020) and by curriculum learning (Bengio et al., 2009), as we detail next.

### 3.2.2 POSITIVE-ONLY LEARNING

The Poincaré distance is minimized when embeddings $h$ and $\hat{h}$ match, both in terms of angle (cosine distance) and radius (uncertainty). Matching uncertainty of two embeddings only really makes sense for positive pairs. In fact, two views of the same sample may well be required to be known to the same degree. However, repulsing negatives in terms uncertainty is not really justifiable, as the sample uncertainty is detached from the sample identity. This has not been considered by prior work (Ermolov et al., 2022), but it matters experimentally, as we show in Sec. 4.3.
We leverage BYOL (Grill et al., 2020) and learn from positive pairs only. Referring to Fig. 1*(b)*, we add a predictor head $q$ to the online branch, to yield the predicted output embedding $p$. Next, $p$ is mapped into the hyperbolic space to get the embedding $z$ which is pushed to match the target embedding $\hat{z}$, reference from the stop-gradient branch, using Eq. 3. Note that this simplifies the model and training, as it removes the need for a memory bank and complex negative mining procedures.

### 3.2.3 CURRICULUM LEARNING

The model random initialization yields starting (high-dimensional) embeddings $h$ and $\hat{h}$ with random directions and high norms, which results in vanishing gradients, cf. (Guo et al., 2022b) and the former discussion on training dynamics. We draw inspiration from curriculum learning (Bengio et al., 2009) and propose to initially optimize for the embedding angles only, neglecting uncertainties. The initial training loss is therefore the cosine distance of $h$ and $\hat{h}$. Since the hyperbolic space is conformal with the Euclidean, the hyperbolic and Euclidean cosine distances coincide:

$$L_{cos}(h, \hat{h}) = \frac{h \cdot \hat{h}}{\|h\|\|\hat{h}\|} \tag{6}$$

The curriculum procedure may be written as the following convex combination of losses:

$$L_{\text{HYSP}}(h, \hat{h}) = \alpha \, L_{poin}(h, \hat{h}) + (1 - \alpha) \, L_{cos}(h, \hat{h}) \tag{7}$$

whereby the weighting term $\alpha$ makes the optimization to smoothly transition from the angle-only to the full hyperbolic objective leveraging angles and uncertainty, after an initial stage. (See Eq. 8 in Appendix A).

---

[§]The reported MCC results refer to the ST-GCN encoder, for a fair comparison, i.e. this is adopted by all methods in the table.

Table 1: Results of linear, semi-supervised and finetuning protocols on NTU-60 and NTU-120.

| | Linear eval. | | | | Semi-sup. (10%) | | Finetune | | | | Additional Techniques | | | | |
|---|---|---|---|---|---|---|---|---|---|---|---|---|---|---|---|
| | NTU-60 | | NTU-120 | | NTU-60 | | NTU-60 | | NTU-120 | | | | Extra | Extra | |
| *Method* | *xsub* | *xview* | *xsub* | *xset* | *xsub* | *xview* | *xsub* | *xview* | *xsub* | *xset* | *3s* | *Neg.* | *Aug.* | *Pos.* | *ME* |
| P&C *Su et al. (2020)* | 50.7 | 76.3 | 42.7 | 41.7 | - | - | - | - | - | - | | | | | |
| MS²L *Lin et al. (2020b)* | 52.6 | - | - | - | 65.2 | - | 78.6 | - | - | - | | ✓ | | | |
| AS-CAL *Rao et al. (2021)* | 58.5 | 64.8 | 48.6 | 49.2 | - | - | - | - | - | - | | ✓ | | | |
| SkeletonCLR *Li et al. (2021)* | 68.3 | 76.4 | 56.8 | 55.9 | 66.9 | 67.6 | 80.5 | 90.3 | 75.4 | 75.9 | | ✓ | | | |
| MCC§ *Su et al. (2021)* | - | - | - | - | 55.6 | 59.9 | 83.0 | 89.7 | 79.4 | 80.8 | | ✓ | | | |
| AimCLR *Guo et al. (2022a)* | 74.3 | 79.7 | 63.4 | 63.4 | - | - | - | - | - | - | | ✓ | ✓ | ✓ | |
| ISC *Thoker et al. (2021)* | 76.3 | **85.2** | **67.9** | **67.1** | 65.9 | 72.5 | - | - | - | - | | ✓ | ✓ | | ✓ |
| **HYSP** *(ours)* | **78.2** | 82.6 | 61.8 | 64.6 | **76.2** | **80.4** | **86.5** | **93.5** | **81.4** | **82.0** | | ✓ | | | |
| 3s-ST-GCN | - | - | - | - | - | - | 85.2 | 91.4 | 77.2 | 77.1 | ✓ | | | | |
| 3s-SkeletonCLR *Li et al. (2021)* | 75.0 | 79.8 | - | - | - | - | - | - | - | - | ✓ | ✓ | | | |
| 3s-Colorization *Yang et al. (2021)* | 75.2 | 83.1 | - | - | 71.7 | 78.9 | 88.0 | 94.9 | - | - | ✓ | | | | |
| 3s-CrosSCLR *Li et al. (2021)* | 77.8 | 83.4 | 67.9 | 66.7 | 74.4 | 77.8 | 86.2 | 92.5 | 80.5 | 80.4 | ✓ | ✓ | | ✓ | |
| 3s-AimCLR *Guo et al. (2022a)* | 78.9 | 83.8 | **68.2** | **68.8** | 78.2 | 81.6 | 86.9 | 92.8 | 80.1 | 80.9 | ✓ | ✓ | ✓ | ✓ | |
| **3s-HYSP** *(ours)* | **79.1** | **85.2** | 64.5 | 67.3 | **80.5** | **85.4** | **89.1** | **95.2** | **84.5** | **86.3** | ✓ | | ✓ | | |

Table 2: Results of linear, semi-supervised and finetuning protocols on PKU-MMD I

| | Linear eval. | | | | Semi-sup. (10%) | | | | Finetune | | | |
|---|---|---|---|---|---|---|---|---|---|---|---|---|
| Method | *Joint* | *Bone* | *Motion* | *3s* | *Joint* | *Bone* | *Motion* | *3s* | *Joint* | *Bone* | *Motion* | *3s* |
| MS²L *Lin et al. (2020b)* | 64.9 | - | - | - | 70.3 | - | - | - | 85.2 | - | - | - |
| SkeletonCLR *Li et al. (2021)* | 80.9 | 72.6 | 63.4 | 85.3 | - | - | - | - | - | - | - | - |
| ISC *Thoker et al. (2021)* | 80.9 | - | - | - | 72.1 | - | - | - | - | - | - | - |
| AimCLR *Guo et al. (2022a)* | 83.4 | 82.0 | **72.0** | 87.8 | - | - | - | 86.1 | - | - | - | - |
| **HYSP** *(ours)* | **83.8** | **87.2** | 70.5 | **88.8** | **85.0** | **87.0** | **77.8** | **88.7** | **94.0** | **94.9** | **91.2** | **96.2** |

## 4 RESULTS

Here we present the reference benchmarks (Sec. 4.1) and the comparison with the baseline and the state-of-the-art (Sec. 4.2); finally we present the ablation study (Sec. 4.3) and insights into the learnt skeleton-based action representations (Sec. 4.4).

### 4.1 DATASETS AND METRICS

We consider three established datasets:

**NTU RGB+D 60 Dataset** (Shahroudy et al., 2016). This contains 56,578 video sequences divided into 60 action classes, captured with three concurrent Kinect V2 cameras from 40 distinct subjects. The dataset follows two evaluation protocols: cross-subject (*xsub*), where the subjects are split evenly into train and test sets, and cross-view (*xview*), where the samples of one camera are used for testing while the others for training.

**NTU RGB+D 120 Dataset** (Liu et al., 2020). This is an extension of the NTU RGB+D 60 dataset with additional 60 action classes for a total of 120 classes and 113,945 video sequences. The samples have been captured in 32 different setups from 106 distinct subjects. The dataset has two evaluation protocols: cross-subject (*xsub*), same as NTU-60, and cross-setup (*xset*), where the samples with even setup ids are used for training, and those with odd ids are used for testing.

**PKU-MMD I** (Chunhui et al., 2017). This contains 1076 long video sequences of 51 action categories, performed by 66 subjects in three camera views using Kinect v2 sensor. The dataset is evaluated using the cross-subject protocol, where the total subjects are split into train and test groups consisting of 57 and 9 subjects, respectively.

The performance of the encoder $f$ is assessed following the same protocols as Li et al. (2021):

**Linear Evaluation Protocol.** A linear classifier composed by a fully-connected layer and softmax is trained supervisedly on top of the frozen pre-trained encoder.
**Semi-supervised Protocol.** Encoder and linear classifier are finetuned with 10% of the labeled data.
**Finetune Protocol.** Encoder and linear classifier are finetuned using the whole labeled dataset.

Table 3: Ablation study on NTU-60 (*xview*). Top-1 accuracy (%) is reported.

| HYSP | w/ neg. | w/o hyper. | w/o curr. learn. | w/ 3-stream ensemble | hyper. downstream |
|------|---------|------------|------------------|----------------------|-------------------|
| 82.6 | 69.7    | 76.9       | 73.9             | 85.2                 | 75.5              |

## 4.2 COMPARISON TO THE STATE OF THE ART

For all experiments, we used the following setup: ST-GCN (Yu et al., 2018) as encoder, Riemannian-SGD (Kochurov et al., 2020) as optimizer, BYOL (Grill et al., 2020) (instead of MoCo) as self-supervised learning framework, data augmentation pipeline from (Guo et al., 2022a) (See Appendix B for all the implementation details).

**NTU-60/120.** In Table 1, we gather results of most recent SSL skeleton-based action recognition methods on NTU-60 and NTU-120 datasets for all evaluation protocols. We mark in the table the additional engineering techniques that the methods adopt: (*3s*) three stream, using joint, bone and motion combined; (*Neg.*) negative samples with or without memory bank; (*Extra Aug.*) extra extreme augmentations compared to *shear* and *crop*; (*Extra Pos.*) extra positive pairs, typically via nearest-neighbor or cross-view mining; (*ME*) multiple encoders, such as GCN and BiGRU. The upper section of the table reports methods that leverage only information from one stream (joint, bone or motion), while the methods in the lower section use 3-stream information.

As shown in the table, on NTU-60 and NTU-120 datasets using linear evaluation, HYSP outperforms the baseline SkeletonCLR and performs competitive compared to other state-of-the-art approaches. Next, we evaluate HYSP on the NTU-60 dataset using semi-supervised evaluation. As shown, HYSP outperforms the baseline SkeletonCLR by a large margin in both *xsub* and *xview*. Furthermore, under this evaluation, HYSP surpasses all previous approaches, setting a new state-of-the-art. Finally, we finetune HYSP on both NTU-60 and NTU-120 datasets and compare its performance to the relevant approaches. As shown in the table, with the one stream information, HYSP outperforms the baseline SkeletonCLR as well as all competing approaches.

Next, we combine information from all 3 streams (joint, bone, motion) and compare results to the relevant methods in the lower section of Table 1. As shown, on NTU-60 dataset using linear evaluation, our 3s-HYSP outperforms the baseline 3s-SkeletonCLR and performs competitive to the current best method 3s-AimCLR. Furthermore, with semi-supervised and fine-tuned evaluation, our proposed 3s-HYSP sets a new state-of-the-art on both the NTU-60 and NTU-120 datasets, surpassing the current best 3s-AimCLR.

**PKU-MMD I.** In Table 2, we compare HYSP to other recent approaches on the PKU MMD I dataset. The trends confirm the quality of HYSP, which achieves state-of-the-art results under all three evaluation protocols.

## 4.3 ABLATION STUDY

Building up HYSP from SkeletonCLR is challenging, since all model parts are required for the hyperbolic self-paced model to converge and perform well. We detail this in Table 3 using the linear evaluation on NTU-60 *xview*:

*w/ neg.* Considering the additional repulsive force of negatives (i.e, replacing BYOL with MoCo) turns HYSP into a contrastive learning technique. Its performance drops by 12.9%, from 82.6 to 69.7. We explain this as due to negative repulsion being ill-posed when using uncertainty, cf. Sec.3.2.

*w/o hyper.* Removing the hyperbolic mapping, which implicitly takes away the self-pacing learning, causes a significant drop in performance, by 5.7%. In this case the loss is the cosine distance, and each target embedding weights therefore equally (no self-pacing). This speaks for the importance of the proposed hyperbolic self-pacing.

*w/o curr. learn.* Adopting hyperbolic self-pacing from the very start of training makes the model more unstable and leads to lower performance in evaluation (73.9%), re-stating the need to only consider angles at the initial stages.

*w/ 3-stream ensemble.* Ensembling the 3-stream information from joint, motion and bone, HYSP improves to 85.2%.

Note that all downstream tasks take the SSL-trained encoder $f$, add to it a linear classifier and train

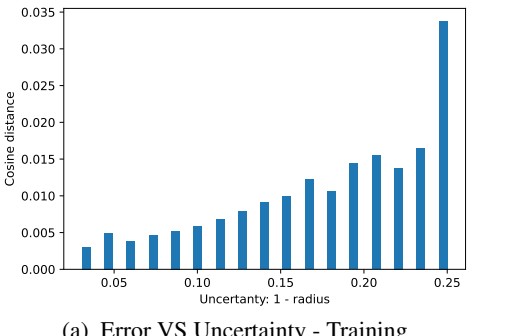

(a)  Error VS Uncertainty - Training

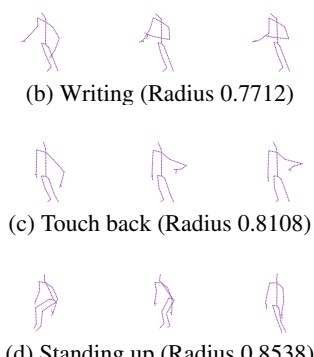

(b) Writing (Radius 0.7712)

(c) Touch back (Radius 0.8108)

(d) Standing up (Radius 0.8538)

Figure 3: (a) Bar plot of the average cosine distance among positive pairs for specific intervals of uncertainty. The plot shows how cosine distance between embeddings and prediction uncertainty after the pre-text task are highly correlated. (b, c, d) Visualizations of skeleton samples. Uncertainty indicates the difficulty in classifying specific actions if they are characterized by less or more peculiar movements (a) and (b) or unambiguous actions (c).

with cross-entropy, as an Euclidean network. Since HYSP pre-trains in hyperbolic space, it is natural to ask whether hyperbolic helps downstream too. A hyperbolic downstream task (linear classifier followed by exponential mapping) yields 75.5%. This is still comparable to other approaches, but below the Euclidean downstream transfer. We see two reasons for this: **i.** in the downstream tasks the GT is given, without uncertainty; **ii.** hyperbolic self-paced learning is most important in pre-training, i.e. the model has supposedly surpassed the information bottleneck (Achille & Soatto, 2018) upon concluding pre-training, for which a self-paced finetuning is not anymore as effective.

### 4.4    MORE ON HYPERBOLIC UNCERTAINTY

We analyze the hyperbolic uncertainty upon the SSL self-paced pre-training, we qualitatively relate it to the classes, and illustrate motion pictograms, on the NTU-60 dataset.

**Uncertainty as a measure of difficulty.** We show in Figure 3a, the average cosine distance between positive pairs against different intervals of the average uncertainty $(1 - \|h\|)$. The plot shows a clear trend, revealing that the model attributes higher uncertainty to samples which are likely more difficult, i.e. to samples with larger cosine distance.

**Inter-class variability.** We investigate how uncertainty varies among the 60 action classes, by ranking them using median radius of the class sample embeddings. We observe that: i) actions characterized by very small movements e.g writing, lie at the very bottom of the list, with the smallest radii; ii) actions that are comparatively more specific but still with some ambiguous movements (e.g. touch-back) are almost in the middle, with relatively larger radii; iii) the most peculiar and understandable actions, either simpler (standing up) or characterized by larger unambiguous movements (handshake), lie at the top of the list with large radii. (See Appendix A for complete list).

**Sample action pictograms** Figure 3 shows pictograms samples from three selected action classes and their class median hyperbolic uncertainty: writing (3b), touch back (3c), and standing up (3d). "Standing up" features a more evident motion and correspondingly the largest radius; by contrast, "writing" shows little motion and the lowest radius.

## 5    LIMITATIONS AND CONCLUSIONS

This work has proposed the first hyperbolic self-paced model HYSP, which re-interprets self-pacing with self-regulating estimates of the uncertainty for each sample. Both the model design and the self-paced training are the result of thorough considerations on the hyperbolic space.

As current limitations, we have not explored the calibration of uncertainty, nor tested HYSP for the image classification task, because modeling has been tailored to the skeleton-based action recognition task and the baseline SkeletonCLR, which yields comparatively faster development cycles.

Exploring the overfitting of HYSP at the latest stage of training lies for future work. This is now an active research topic (Ermolov et al., 2022; Guo et al., 2021; 2022b) for hyperbolic spaces.

## 6 ETHICS STATEMENT

The use of skeletal data respects the privacy of the pictured actors, to a large extent. In most cases, privacy is affected when the actor's appearance is required, from the identity could be inferred. Although it might be possible to recognize people from their motion (e.g. recognition of their Gait), this is not straightforward. This makes a potential misuse case for this work, however less likely, compared to the use of images.

## 7 REPRODUCIBILITY STATEMENT

To ensure the complete reproducibility of our work, we have included in the abstract the link to the official github repository of the paper. The README file contains instructions on how to download the data, preprocess it, train the encoder self-supervisedly, and then evaluate the model on the various protocols described in Sec. 4.1. The datasets are not included in the repository because they are publicly available for download and use. Furthermore, Appendix B provides detailed information about implementation details.

## 8 ACKNOWLEDGEMENT

We are most grateful for the funding from the Panasonic Corporation and the fruitful research discussions with the Panasonic PRDCA-AI Lab. We would like to acknowledge the support of Nvidia AI Technology Center (NVAITC) for computing resources and of Giuseppe Fiameni for his expertise on that, which enabled their effective use. Authors are also grateful to Anil Keshwani for initial discussions on contrastive learning for videos.

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

## A APPENDIX

Here we provide additional visualizations and analysis of the proposed model HYSP. First, we present an analysis of the hyperbolic uncertainty, showing how it varies among different action classes; then, we illustrate how it varies within the same action class with sample pictograms; finally, we illustrate the relation between action variability and prediction error.

**Inter-class variability** In support of the discussion in Sec. 4.5 in the main paper, in Fig. 4 we present the complete list of the 60 action classes from the NTU-60 dataset, ordered according to the median radius of their embeddings[¶]. Note that the radius serves as a valid ranking metric for the median action uncertainty, however this work has not explored the calibration of the actual uncertainty value (cf. the discussion on limitations in Sec. 5). Note in the figure how the radius ranks well the level of peculiarity and unambiguous motion in the sequence, e.g. the pictured "Pushing" action (with radius 0.9697) is easier to recognize than "Shake head" (with radius 0.7181).

In Fig. 4, actions with smaller movements (e.g. writing, eat meal, playing with phone, or brushing teeth) have smallest radii, while actions with comparatively wider movements (e.g. touch-back, hopping or wearing on glasses) are almost in the middle with relatively larger radii. Finally, actions with the most peculiar and unambiguous movements, either simpler (standing up, throw) or involving two people (handshake, walking apart), lie at the top of the list with large radii. Fig. 5 shows the pictograms of some selected action classes. For each action, we have selected a representative sequence (with its radius close to the median of the class).

**Intra-class variability** We complement the above paragraph with qualitative examples of intra-class variability, i.e. how the action embedding radius (as a proxy to its uncertainty) varies for sequences within the same class.

As illustrated in Fig. 6, with reference to actions "Touch neck" (a, b, c) and "Staggering" (d, e, f), some sequences are more ambiguous (6a, 6d), some are a little bit clearer (6b, 6e) and, finally, others are more specific and easier to represent by the model (6c, 6f).

---

[¶]The median is adopted (instead of the mean) because it is a more robust estimator, less affected by outlier values of radii.

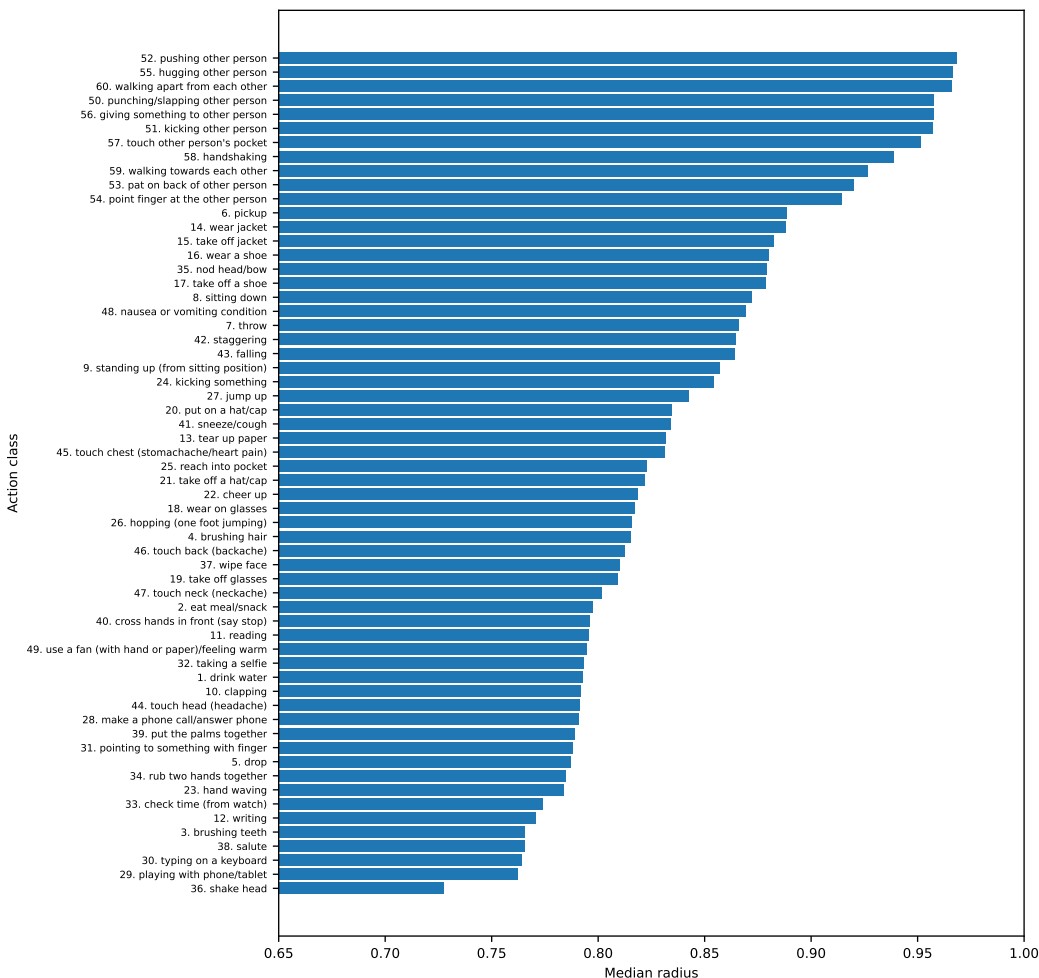

Figure 4: Median radius per action class is an indicator of the understandability of each action class (vector graphics, please zoom-in for better readability).

- 0. drink water.
- 1. eat meal/snack.
- 2. brushing teeth.
- 3. brushing hair.
- 4. drop.
- 5. pickup.
- 6. throw.
- 7. sitting down.
- 8. standing up (from sitting position).
- 9. clapping.
- 10. reading.
- 11. writing.
- 12. tear up paper.
- 13. wear jacket.
- 14. take off jacket.
- 15. wear a shoe.
- 16. take off a shoe.
- 17. wear on glasses.
- 18. take off glasses.
- 19. put on a hat/cap.

- 20. take off a hat/cap.
- 21. cheer up.
- 22. hand waving.
- 23. kicking something.
- 24. reach into pocket.
- 25. hopping (one foot jumping).
- 26. jump up.
- 27. make a phone call/answer phone.
- 28. playing with phone/tablet.
- 29. typing on a keyboard.
- 30. pointing to something with finger.
- 31. taking a selfie.
- 32. check time (from watch).
- 33. rub two hands together.
- 34. nod head/bow.

- 35. shake head.
- 36. wipe face.
- 37. salute.
- 38. put the palms together.
- 39. cross hands in front (say stop).
- 30. sneeze/cough.
- 41. staggering.
- 42. falling.
- 43. touch head (headache).
- 44. touch chest (stomachache/heart pain).
- 45. touch back (backache).
- 46. touch neck (neckache).
- 47. nausea or vomiting condition.
- 48. use a fan (with hand or paper)/feeling warm.

- 49. punching/slapping other person.
- 50. kicking other person.
- 51. pushing other person.
- 52. pat on back of other person.
- 53. point finger at the other person.
- 54. hugging other person.
- 55. giving something to other person.
- 56. touch other person's pocket.
- 57. handshaking.
- 58. walking towards each other.
- 59. walking apart from each other.

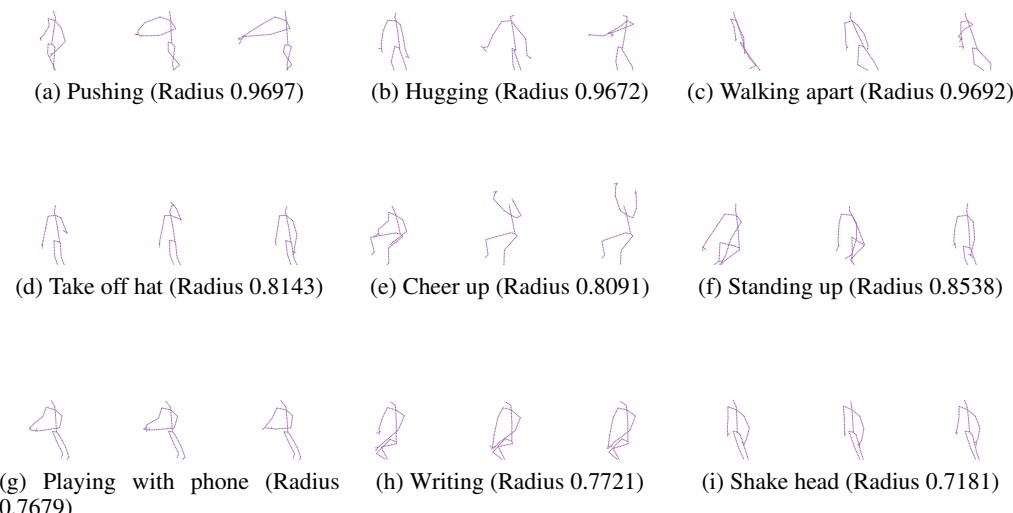

Figure 5: Visualization of inter-class variability through skeleton sequences with high (a,b,c), average (d,e,f), and low (g,h,i) median radius.

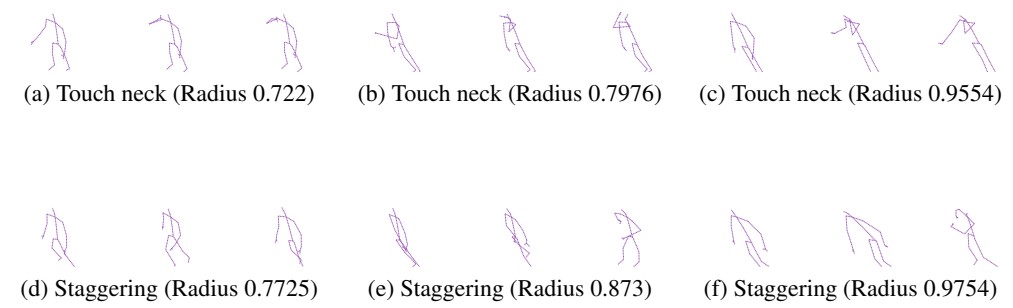

Figure 6: Visualization of intra-class variability through skeleton sequences of the same class having low (a,d), average (b,e), and high (c,f) radius.

**Relation between intra-class variability and prediction error**    We complement the study on action variability and uncertainty by relating those to the prediction error. In Fig. 7, we report the confusion matrix, whereby actions have been arranged by the median radius of the class embeddings. So, actions in the rows and columns are sorted according to their median radii (increasing left-to-right and top-to-bottom), using the values from Fig. 4. One observes that lower radii (bottom right corner) yield larger prediction errors among the actions. This is the case for actions such as writing (#11) and brushing teeth (#2), i.e. these are more ambiguous/uncertain because they are characterized by less distinct motion. (The action IDs are illustrated after the confusion matrix.)

## B    APPENDIX

**Transfer Learning protocol**    Following  Thoker et al. (2021), we additionally test models using the transfer learning protocol. We leverage the pre-trained models on the NTU-60 and PKU-MMD I datasets and finetune them (pre-trained encoder and a classifier jointly) on PKU-MMD II. HYSP outperforms all competing methods in both transfers. Results are reported in Table 4.

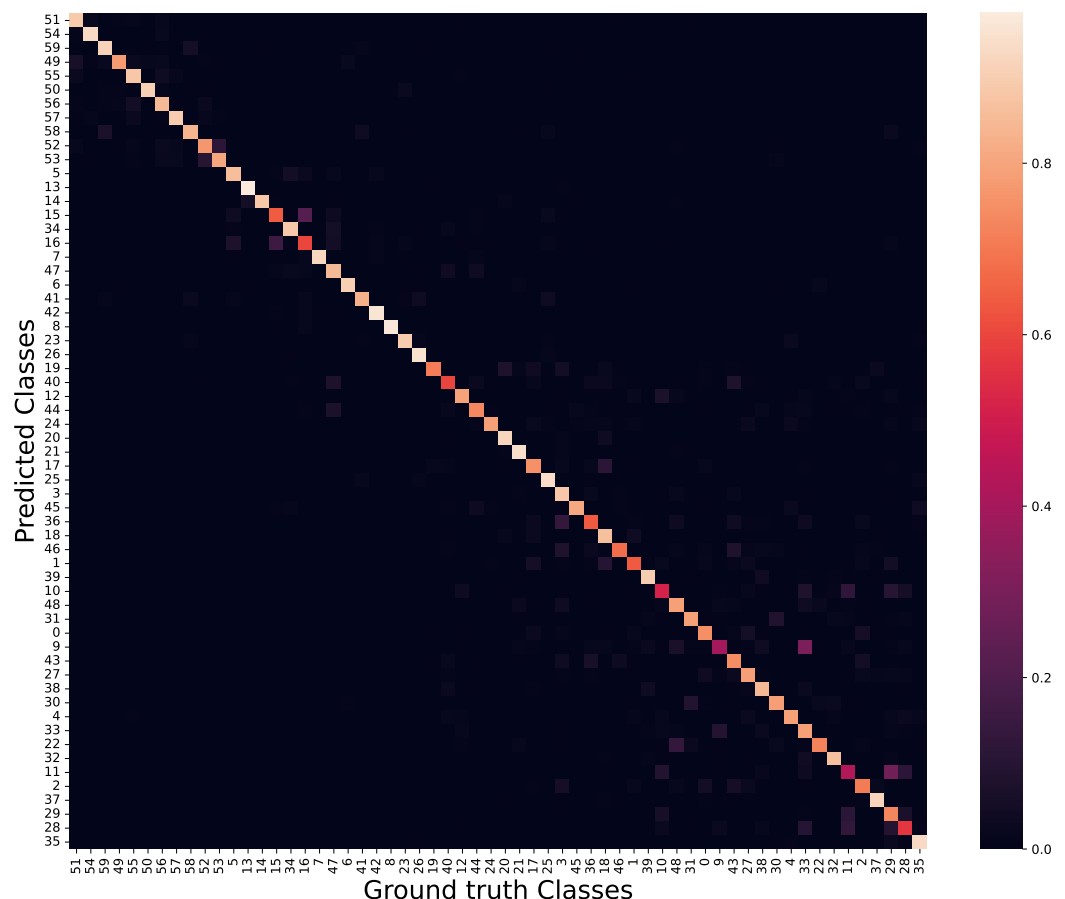

Figure 7: Confusion matrix among the actions of NTU60 (*xview*, single stream input, linear test protocol, top-1 accuracy) arranged by decreasing order of the median radii of their class sample embeddings (values from Fig. 4). Actions with lower radii (bottom-right corner) result in larger prediction errors. Those actions (IDs listed above) are also more ambiguous and characterized by less distinct motions.

Table 4: Results of transfer learning on PKU-MMD II, comparing the proposed HYSP against more recent techniques (performance scores other than HYSP are reported from Thoker et al. (2021)). The evaluation protocol follows Thoker et al. (2021): after training an encoder on the source dataset (NTU-60 and PKU-MMD I), the pre-trained encoder and classifier are jointly fine-tuned on a target dataset (PKU-MMD II) for action classification.

| | Transfer Learning (PKU-MMD II) | |
| --- | --- | --- |
| Method | *PKU-MMD I* | *NTU-60* |
| S+P *Zheng et al. (2018)* | 43.6 | 44.8 |
| MS$^2$L *Lin et al. (2020b)* | 44.1 | 45.8 |
| ISC *Thoker et al. (2021)* | 45.1 | 45.9 |
| **HYSP** *(ours)* | **50.7** | **46.3** |

**Data pre-processing** Following Li et al. (2021), we pre-process each skeleton sequence by removing all invalid frames and resizing it to the length of 50 frames by linear interpolation. We use two sets of augmentations, normal and extreme. *Normal augmentations* include *Shear* (Rao et al., 2021) as the spatial and *Crop* (Shorten & Khoshgoftaar, 2019) as the temporal augmentation. *Extreme augmentations* (Guo et al., 2022a) include four spatial: *Shear, Spatial Flip, Rotate, Axis Mask*, two temporal: *Crop, Temporal Flip*, and two spatio-temporal augmentations: *Gaussian Noise and*

Table 5: Top-1 accuracy varying batch size for the linear protocol. The dataset used in this table is NTU-60 xview.

| Batch size | 512 | 256 | 128 | 64 | **32** |
|---|---|---|---|---|---|
| Top-1 Acc. | 78.3 | 79.9 | 81.2 | 82.2 | **82.6** |

*Gaussian Blur.* Normal augmentations are applied on the target, while extreme ones are applied on the online sequences.

**Self-supervised Pre-training**  The encoder $f$ is ST-GCN (Yu et al., 2018) with output dimension 1024. Following BYOL (Grill et al., 2020), the projector and predictor MLPs are linear layers with dimension 1024, followed by batch normalization, ReLU and a final linear layer with dimension 1024. The model is trained with batch size 512 and learning rate $lr$ 0.2 in combination with RiemannianSGD (Kochurov et al., 2020) optimizer with momentum 0.9 and weight decay 0.0001. For curriculum learning, across all experiments, we set $e_1 = 50$ and $e_2 = 100$ in Eq. 8.

$$\alpha(e) = \begin{cases} 0 & \text{if } e \leq e_1 \\ \frac{e-e_1}{e_2-e_1} & \text{if } e_1 < e < e_2 \\ 1 & \text{if } e \geq e_2 \end{cases} \tag{8}$$

Training on 4 Nvidia Tesla A100 GPUs takes approximately 8 hours.

**Evaluation**  In downstream evaluation, the model is trained for 100 epochs using SGD optimizer with momentum 0.9 and weight decay 0. For the linear evaluation, the base $lr$ of the classifier is set to 10, dropped by a factor 10 at epochs 60 and 80, keeping the encoder frozen. For semi-supervised and fine-tuned evaluation, the base $lr$ of the encoder and classifier is set to 0.1, dropped by a factor 10 at epochs 60 and 80. Exponential mapping is not adopted in evaluation phase.

## C  APPENDIX

**Effect of batch size on linear protocol**  Here we report an interesting finding relating to the batch-size hyper-parameter when testing with the linear protocol. We find that a lower batch size (32) results in a considerable performance increase with respect to a larger one (512). For the NTU-60, *xview*-subset, the top-1 accuracy grows from 78.3 to 82.6, resulting in a performance increase of +4.3%. For this test, Table 5 reports performances of various batch sizes.

**Effect of higher/lower uncertainty samples on training**  We experiment to compare the performance variation between using harder positives only (i.e. the samples with higher uncertainty) Vs. easier positives only (those samples with lower uncertainty). For the sake of comparison, we select an equal number of hard and easy positives, namely the first and second half of the total samples, after sorting them in decreasing order of uncertainty (as estimated upon the HYSP pre-training on all samples). It turns out that learning with the harder examples yields the top-1 accuracy of 70.9%, while the easier ones yield 74.5%. It is expected that both results are well below the performance of the model trained on the full dataset (82.6% with the joint stream), because of using half the amount of data. As for the performance discrepancy, we speculate that easier samples have an edge on the half-sized training set, while harder samples better support when larger amount of data is available. Let us also add that, as expected, the two new trainings have different durations: that one with harder samples converges and overfits earlier (at epochs 400 and 500 respectively) than with easier (epochs 400 and 700 correspondingly).

## D  APPENDIX

**Additional insights into the self-paced learning**  In Fig. 8, we provide further insights into Eq. 5, the Riemannian gradient of the Poincaré loss, by illustrating the relation between the gradient magnitude and the uncertainty of the target-view embedding $(1 - \|\hat{h}\|)$. We consider the model checkpoint

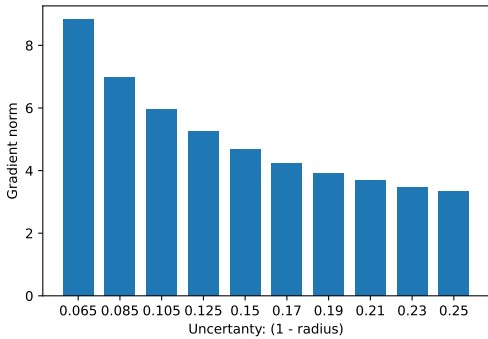

Figure 8: Bar plot of the average Riemannian gradient norm for specific intervals of uncertainty. The plot experimentally explains the intuition behind Eq. 5 described in Sec 3.2.1. The gradient norm and prediction uncertainty after the pre-text task are highly correlated. Learning them is *self-paced*, because lower the uncertainty of $\hat{h}$ and the stronger is the gradient.

for epoch 100, which belongs to the second (the main) stage of training (the analysis of other epochs has produced similar statistics). Then we compute aggregate histogram statistics across all the training samples. The plot of Fig. 8 confirms the analysis of Sec. 3.2.1: the lower the uncertainty of the target view embedding $\hat{h}$ (left side of the plot), the larger the gradient. As a result, learning is self-paced because the importance of each sample is weighted, via the respective gradient magnitude, according to the level of certainty of the target-view embedding.

