# OpenReview forum: "Hyperbolic Self-paced Learning for Self-supervised Skeleton-based Action Representations"
_ICLR.cc/2023/Conference — ICLR 2023 poster_

### Official Review · Reviewer_GGZs · 2022-10-17

**Confidence:** 3
**Correctness:** 3
**Technical Novelty And Significance:** 3
**Empirical Novelty And Significance:** 3
**Recommendation:** 6

**Clarity, Quality, Novelty And Reproducibility:**

- Clarity and Quality: The paper is well organized.
- Novelty: The contributions are somewhat new. Aspects of the contributions exist in prior work.
- Reproducibility: The author provides the code. There is reason to believe that the paper is reproducible.

**Strength And Weaknesses:**

- Strength:
    - The paper is well written and the problem being tackled is clearly described.
    - The introduction of hyperbolic mapping and curriculum learning brings a different perspective to self-supervised skeleton action recognition.
    - The paper has a detailed theoretical basis and the detailed visible analysis (Figure 2,4,5) further makes the idea more convincing.

- Weakness:
    - Parts of the approach do exist in prior works. The authors should further elaborate whether the HYSP is a combination of several methods (BYOL + Ganea et al. 2018 + Guo et al. 2022). However, I think that the paper presents a good study on some of the practices and makes a compelling case for using BYOL for the task.
    - HYSP is not the first method to use the positive-only BYOL and the performance of HYSP is not that competitive. *''Moliner et al. Bootstrapped Representation Learning for Skeleton-Based Action Recognition. In Proceedings of the IEEE/CVF Conference on Computer Vision and Pattern Recognition (CVPR) Workshops, 2022, pp. 4154-4164.''*
    - Although mentioned in the conclusion, I still think that HYSP is more like a general improvement of the self-supervised framework for BYOL combined with hyperbolic space. Measurements on image classification tasks may be more convincing and necessary. I have not seen a unique improvement and analysis for the field of skeleton action recognition.

- Other questions:
    - Double check the last name and first name of the author of the cited paper, some are confused, such as "Tianyu et al." should be "Guo et al.".
    - Please explain Eq.5 in more detail. The conclusion behind Eq.5 cannot be intuitively understood from Eq.5.
    - In Table 1, SkeletonCLR also uses negative samples.
    - Does "w/o hyper." in Table 3 mean using BYOL directly? If yes, I think the accuracy is unsatisfactory (see weekness2). If not, please explain in more detail and give the result of using BYOL directly.


**Summary Of The Paper:**

While most works in self-supervised skeleton-based action recognition have used MoCo for modeling, this paper makes a case for using the positives-only design of BYOL instead. The paper proposes the HYperbolic Self-Paced model (HYSP) according to the principle that more certain samples should drive the learning process more predominantly. The authors empirically demonstrate that the proposed approach achieves superior performance on public datasets.

**Summary Of The Review:**

The paper provides a new framework to model self-supervised skeleton-based action recognition. The paper provides extensive experiments for that and interesting observations and analyses showing the advantages of the proposed method. There are a few potential issues detailed in the previous section. Considering all the strengths and weaknesses pointed out in the Main review, I give the current rating.

---

> ### Author Response · Authors · 2022-11-18
> **Response to Reviewer 4**
>
> 1. Thanks for the comment and appreciation.
> Indeed, the mentioned methods are instrumental model choices: hyperbolic geometry [1] has been chosen as it provides uncertainty; BYOL [2] has been selected because matching the uncertainty of views is mainly justifiable for positives (cf. Sec. 3.2.2); and extra augmentations [3] boost performance of skeleton-based action recognition.
> The claimed contribution of HYSP sets a target beyond the mere combination of methods, since no earlier method self-paced SSL learning by hyperbolic uncertainty.
>
> 2. Thanks for the reference, which we have added to the paper and discussed in related work (Sec. 2.4).
> [4] is effectively the first skeleton-based SSL using the positive-only BYOL. We have amended the statements in the paper.
> The main claim of HYSP is: "the first hyperbolic self-paced learning model for SSL'' (Sec. 1), and self-pacing with hyperbolic uncertainty remains a novelty w.r.t. [4].
> With regard to the performance, we have noted a few differences in the data preparation protocol, most notably the length of the input sequences, since [4] crops longer excerpts. This differs from all other techniques in Tables 1, 2, and 5. We note that their code is not currently available, but we asked for the code and data preparation protocol, to estimate comparable figures. We will add the comparison to the paper as soon as available.
>
> 3. Thanks for your consideration. Please consider that hyperbolic self-paced SSL is novel, to the best of our knowledge.
> The choice of hyperbolic geometry is instrumental, as it provides uncertainty, as means to self-pace. Similarly, the choice of BYOL is instrumental, as matching the uncertainty for positives is only really justifiable.
> Please note that none of the most recent papers on skeleton-based SSL (cf. the list of methods in Tables 2, 3, 5 of the paper) test image classification tasks.
> We appreciate the reviewer's statement that HYSP provides general improvements on SSL, and we agree that research should address this. Still, we maintain that this extension is more suitable for follow-up works or journal extensions.
>
> 4. Thank you for the note, we have corrected it and verified all other references.
>
> 5. For a better intuition, beyond the explanation of Sec. 3.2.1, in Fig. 8 of Appendix D, we have provided further insights into Eq. 5, the Riemannian gradient of the Poincaré loss, by illustrating the relation between the gradient magnitude and the uncertainty of the target-view embedding $(1 - \|\hat{h}\|)$.
> We consider the model checkpoint for epoch 100, which belongs to the second (the main) stage of training (the analysis of other epochs has produced similar statistics). Then we computed aggregate histogram statistics across all the training samples.
> The plot of Fig. 8 confirms the analysis of Sec. 3.2.1: the lower the uncertainty of the target view embedding $\hat{h}$ (left side of the plot), the larger the gradient. As a result, learning is self-paced because the importance of each sample is weighted, via the respective gradient magnitude, according to the level of certainty of the target-view embedding.
>
> 6. Thanks for the remark; we have added the missing checkmark.
>
> 7. Yes, "w/o hyper'' is BYOL, but there was a typo in the text. We apologize for this inconvenience.
> Upon the batch-size correction (cf. R2.1), the performance of BYOL-only is 76.9 (NTU-60, *xview*), which is better than MoCo-only "w/ neg.'', 69.7, but below the proposed HYSP, 82.6.
> Please see Table 3 in the updated manuscript.
>
> ---
>
> [1] *Hyperbolic neural networks* (Ganea et al. NeurIPS 2018)
> [2] *Bootstrap Your Own Latent: A new approach to self-supervised learning* (Grill et al. NeurIPS 2020)
> [3] *Contrastive Learning from Extremely Augmented Skeleton Sequences for Self-supervised Action Recognition* (Guo et al. AAAI 2022)
> [4] *Bootstrapped Representation Learning for Skeleton-Based Action Recognition* (Moliner et al. CVPR Workshop 2022)

---

### Official Review · Reviewer_V8KY · 2022-10-22

**Confidence:** 5
**Correctness:** 3
**Technical Novelty And Significance:** 3
**Empirical Novelty And Significance:** 2
**Recommendation:** 6

**Clarity, Quality, Novelty And Reproducibility:**

Clarity:
This paper is well-written.

Quality:
The paper has minor technical flaws.

Novelty:
The novelty of the proposed method is marginal

Reproducibility:
The authors have provided the code in the supplemental materials.

**Strength And Weaknesses:**

Pros:

(1) This is the first self-supervised skeleton representation work on hyperbolic space.

(2) This paper is well written.


Cons:

(1) The proposal seems to be incremental based on prior works. HYSP is not the first method that uses the hyperbolic space in skeleton action recognition [1*] and uses the BYOL for self-supervised skeleton representation learning [2*].

(2) Why only BYOL?: In my opinion, a better way to show the hyperbolic space is useful is to combine the hyperbolic space with multiple self-supervised learning baselines. From the results in Table 3 (w/ neg.), the hyperbolic space seems not to work well on MoCo.

(3)  The linear evaluation results are not as good as SOTA methods. And the transfer learning results are missing. These two evaluation protocols are more important than the semi-supervised and supervised settings, regarding showing the effectiveness of self-supervised methods, in my opinion.

(4) Table 3: the BYOL-only result (w/o hyper.) may be not reasonable.




[1*] Mix Dimension in Poincar\'{e} Geometry for 3D Skeleton-based Action Recognition. ACM MM 2020.

[2*] Bootstrapped Representation Learning for Skeleton-Based Action Recognition. CVPRW 2022.

**Summary Of The Paper:**

This paper proposes a self-supervised skeleton representation learning method based on the BYOL and hyperbolic space. The experimental results show that the proposed HYSP model achieves state-of-the-art performances on three public datasets.

**Summary Of The Review:**

As mentioned in the 'Strength And Weaknesses', both the hyperbolic space and BYOL are not new in skeleton action recognition. So I tend to give the 'marginally below the acceptance threshold'.

---

> ### Author Response · Authors · 2022-11-18
> **Response to Reviewer 3**
>
> 1. Thanks for both the references, which have been added to the manuscript and described in related work (Sec. 2.4).
> In defense of our claims of novelty, [1] introduces hyperbolic geometry for the task of supervised skeleton-based action recognition. We claim that HYSP be "the first hyperbolic self-paced learning model for SSL''. Both the self-pacing and the SSL aspects differ substantially from [1].
> Let us further note that [1] leverages hyperbolic geometry for its hierarchical properties, while HYSP leverages it for the hyperbolic uncertainty, also a key difference (cf. also R2.2).
> With respect to [2], we agree with the reviewer that [2] is the first skeleton-based SSL using BYOL. We note that [2] introduces augmentations (e.g. multi-view sampling), which might be complementary to HYSP, instead focussed on self-paced SSL learning with hyperbolic uncertainty. In fact, our main claimed novelty of self-pacing with hyperbolic uncertainty is distant from their work and still applies. Please also see our comment R4.2.
>
> 2. The reviewer makes a fair remark on the potential applicability of the proposed self-pacing hyperbolic to other SSL frameworks.
> At an earlier research stage, we had tested similar model enhancements to SimSiam [7], but achieved inferior performance than BYOL. As this matched image SSL tasks, we did not consider it further.
> As for MoCo, please see our argument in Sec. 3.2.2: "matching uncertainty of two embeddings only really makes sense for positive pairs'', and our note on uncertainty for negatives.
> Integrating self-pacing hyperbolic uncertainty into more recent techniques such as NNCLR [8] and VICReg [9] will involve more research, and it remains currently outside the scope of this work.
>
> 3. **Linear evaluation.** Thanks for your comment. We have checked the submitted results and found that we had neglected to tune the batch-size hyperparameter (cf. the comment to R2.1 and Table 5 in Appendix C). Adjusting it on the NTU-60 validation set, improved the linear evaluation performance of HYSP consistently in all experiments. This makes HYSP the best overall performer (3-stream experiment), and it better reflects the rankings on other evaluation protocols. Tables 1 and 2 in the paper have been updated.
> **Transfer Learning.** Thanks for the recommendation. Following [5], we have tested transfer learning from the NTU-60 and PKU-MMD I sources to PKU-MMD II. It shows from Table 4 in Appendix B that HYSP outperforms the state-of-the-art.
>
> 4. This was a typo, we apologize for this inconvenience.
> Upon the batch-size correction (cf. R2.1), the performance of BYOL-only is 76.9 (NTU-60, *xview*), which is better than MoCo-only, 69.7. Hyperbolic self-pacing improves on BYOL by +5.7\%, reaching 82.6. Table 3 has been updated.
>
> ---
>
> [1] *Mix Dimension in Poincaré Geometry for 3D Skeleton-based Action Recognition* (Peng et al. ACMMM 2020)
> [2] *Bootstrapped Representation Learning for Skeleton-Based Action Recognition* (Moliner et al. CVPR Workshop 2022)
> [3] *MS2L: Multi-Task Self-Supervised Learning for Skeleton Based Action Recognition* (Lin et al. ACMMM 2020)
> [4] *3D Human Action Representation Learning via Cross-View Consistency Pursuit* (Li et al. CVPR 2021)
> [5] *Skeleton-Contrastive 3D Action Representation Learning* (Thoker et al. 2021 ACM Multimedia 2021)
> [6] *Contrastive Learning from Extremely Augmented Skeleton Sequences for Self-supervised Action Recognition* (Guo et al. AAAI 2022)
> [7] *Exploring simple siamese representation learning* (Chen et al. CVPR 2021)
> [8] *With a little help from my friends: Nearest-neighbor contrastive learning of visual representations* (Dwibedi et al. ICCV 2021)
> [9] *Vicreg: Variance-invariance-covariance regularization for self-supervised learning* (Bardes et al. ICLR 2022)

---

### Official Review · Reviewer_R1Eg · 2022-10-23

**Confidence:** 4
**Correctness:** 3
**Technical Novelty And Significance:** 3
**Empirical Novelty And Significance:** 3
**Recommendation:** 6

**Clarity, Quality, Novelty And Reproducibility:**

- The paper is well written and easy to follow for the most part. Please look at the weaknesses for some suggestions on how to improve clarity.
- The approach combines ideas from four different areas in a novel way.

**Strength And Weaknesses:**

Strengths:
- The paper is well written and easy to follow. The intuitions are clear for the most part.
- I like the idea of self-pacing for self-supervised learning. The paper proposes a novel approach based on hyperbolic geometry to tackle this.
- The paper combines ideas from four different areas. The combination is a novel contribution to the field.
- The approach leads to good gains on some of the tasks considered.

Weaknesses:
- While the approach shows good improvements on semi-supervised learning, these do not hold for the other settings. The comparison for finetuning would have been easier if there were numbers reported for ISC.
- It is not clear from the paper why hyperbolic specifically. The Section 3.2, directly starts with explaining how a hyperbolic extension to SkeletonCLR would work. As far as I understand, the hyperbolic geometry is useful to encode hierarchies, tree-like structures. It is not clear to me why the problem being tackled by this paper needs that.
- Useful intuition to clarify : What does it mean for a target to be uncertain. I found an exaplanation to this in caption for Figure 3. I suggest you include this earlier in the text to ease understanding.
- A non-hyperbolic self-paced learning baseline : From Fig 3 caption ".. Uncertainty indicates the difficulty in classifying specific actions if they are characterized by less or more peculiar..". Is it possible to weigh the usual contrastive learning loss using the distance between positives as a baseline to determine uncertainty?
- Given that your approach is based on a different framework compared to state-of-the-art, you should include results of a non-hyperbolic baseline with BYOL. This will help determine how much of the improvement can be obtained using BYOL based baseline vs MoCo,  For example, for Image SSL tasks, BYOL is generally better than MoCo-v2.
- Drop in performance with negatives : While the argument for why negatives cannot be generated in the h space seems convincing, what happens if you generate negatives/apply contrastive learning loss for this experiment on the z space (after projection g) ?
- Suggestion : I think in addition to linear evaluation, you should evaluate your approach on transfer learning as well. For skeleton-based SSL, NTU -> PKU I/II is a good option.
- The paper includes details on training time. It would be helpful to include details on how much extra compute/traning time is needed due to the use of Riemannian gradient descent.


**Summary Of The Paper:**

- The paper explores hyperbolic self paced learning for self supervised learning of action representations.
- The key idea is to use hyperbolic geometry to automatically order the training samples from easy to hard. This determines the learning pace thus improving learning.
- The framework is borrowed from BYOL, which does not need negatives.
- The approach shows promising performance on the tasks considered.

**Summary Of The Review:**

The paper proposes a new technique for SSL on skeleton sequences. The improvements are decent, though not consistent on all tasks. I have some concerns on baselines and intuitions and hence my current score.

---

> ### Author Response · Authors · 2022-11-18
> **Response to Reviewer 2**
>
> 1. Thanks for your comment. This has been the stimulus to further investigate the linear protocol experiments and we have found that a lower batch size (32, instead of 512) results in a considerable performance increase, e.g. +4.3\%, from 78.3 to 82.6, for the NTU-60, *xview*-subset. For this test, the table below reports performances of various batch sizes.
> Results in Tables 1 and 3 of the main paper have been updated, to report top-1 accuracy for linear protocol tests with a batch size of 32. The HYSP performance is now consistent across all protocols, with the exception of NTU-120 linear protocol, for which an explanation remains to address in further research. This study has been added to Appendix C.
> Please note: we do not report ISC [1] fine-tuning results because those are not included in their paper.
> | | | | | | |
> |---|---|---|---|---|---|
> | **_Batch size_** | 512 | 256 | 128 | 64 | **32** |
> | **_Top-1 Acc._** | 78.3 | 79.9 | 81.2 | 82.2 | **82.6** |
>
> 2. The literature is split on the motivation for hyperbolic geometry, and this probably obfuscates the necessity for hyperbolic in this paper. A body of work leverages hyperbolic geometry (mostly the Poincaré model) to encode hierarchies and tree-like structures [2,3]. This leverages the exponentially growing volume towards the Poincaré ball edges. Others adopt hyperbolic geometry (mostly the Poincaré model, too) since it yields a notion of uncertainty for each sample embedding, encoded by its radius in the Poincaré ball [4,5]. This emerges from the usage of the Poincaré distance, which grows exponentially with the radius. So the model should tend to assign less certain matches a lower radius, to alleviate penalization for mismatching embeddings.
> This work stems from the second line of works. HYSP leverages the sample embedding radius, as a proxy for the sample uncertainty, to self-pace learning (more certain samples should drive the training more than less certain ones). The radius estimation is a by-product of the hyperbolic SSL optimization, and the sample radius embeddings evolve during training (they are end-to-end trained with the main positive sample matching objective). We clarify in Sec. 2.2 of the paper.
>
> 3. In agreement with the literature on hyperbolic geometry concerning uncertainty [4, 5] (cf. the previous comment), uncertainty relates to the radius of the sample embedding in the Poincaré ball. In practice, the uncertainty indicates the difficulty of the encoder in representing an action sequence. It is greater if the sequence has confusing or unusual moves, while it is lower if the movements are clear and comprehensible. We include this in Sec. 3.2 of the paper.
>
> 4. Adopting the non-hyperbolic (Euclidean) distance between positives as the optimization self-pace may not yield the desired improvement, because the same distance is also used as loss, in the Euclidean cases.
> The case of the hyperbolic geometry is different because the radius estimation (the proxy to uncertainty, and determining the learning pace in HYSP) is self-regulated by the exponentially growing Poincaré distance (the actual loss).
> (Please see also R4.5 and the novel Appendix D).
> Other works provide non-hyperbolic uncertainty estimates [6,7], but none considers it for SSL. Research on different types of uncertainty in this realm appears of interest, but it has not been considered in this work and it may require dedicated research.
>
> 5. The baseline (non-hyperbolic) BYOL had been included in Table 3, termed "w/o hyper", but there was a typo in the value, which has now been corrected (thanks for the remark).
> The top-1 accuracy (NTU-60, _xview_) of the baseline BYOL (76.9\%) is slightly better than the contrastive counterpart using MoCo (76.4\%), "SkeletonCLR" in Table 2. This is consistent with image SSL tasks. The proposed HYSP further improves the BYOL baseline by 5.7\%.
>
> 6. Thanks for the observation. We have experimented but the model did not converge. In particular, we kept the hyperbolic loss between $h$ and $\hat{h}$, and we added a contrastive loss for the negatives between $z$ (or $p$ after projection) and $\hat{z}$. We speculate that it might be difficult to optimize the model parameters for both the Euclidean and the hyperbolic spaces.
>
> 7. Thanks for the suggestion. For the transfer learning we leveraged the pretrained models on the NTU-60 and PKU-MMD I datasets and finetuned them (pretrained encoder and a classifier jointly) on PKU-MMD II, following [1].
> The results are illustrated in Table 4 in Appendix B, where we compare against those reported in ISC [1]. For both experiments, HYSP outperforms the state-of-the-art.
>
> 8. The difference between the Euclidean and the Riemannian gradients is given by the conformal factor, provided explicitly in Eq. 8 of the paper [8].
> In practice, the training time is not affected by adding the conformal factor and changing from Euclidean to Riemannian gradient descent.

---

> > ### Author Response · Authors · 2022-11-18
> > **References**
> >
> > [1] *Skeleton-Contrastive 3D Action Representation Learning* (Thoker et al. ACM Multimedia 2021)
> > [2] *From trees to continuous embeddings and back: Hyperbolic hierarchical clustering* (Chami et al. NeurIPS 2020)
> > [3] *Hyperbolic image embeddings* (Khurulkov et al. CVPR 2020)
> > [4] *Learning the predictability of the future* (Suris et al. CVPR 2021)
> > [5] *Hyperbolic image segmentation* (Atigh et al. CVPR 2022)
> > [6] *Dropout as a Bayesian Approximation: Representing Model Uncertainty in Deep Learning* (Gal and Ghahramani, NeurIPS 2015)
> > [7] *Self-paced and self-consistent co-training for semi-supervised image segmentation* (Wang et al. Medical Image Analysis 2021)
> > [8] *Clipped hyperbolic classifiers are super-hyperbolic classifiers* (Guo et al. CVPR 2022)

---

> > > ### Comment · Reviewer_R1Eg · 2022-11-23
> > > **Thanks for the response**
> > >
> > > I thank the authors for their detailed response to my concerns.
> > > I found the response to be adequate for the most part. Thanks for trying out the contrastive extension (6). I think that baseline should work but it can be explored as a part of future works. I agree with Reviewer GGZs, the general nature of this work should be explored in the future works as well.
> > >
> > > Given the batch size observation on the linear evaluation protocol, I am curious if it extends to other comparable skeleton-based SSL works. While not necessary, did the authors check this ?
> > >
> > > I have increased my rating.

---

> > > > ### Author Response · Authors · 2022-12-02
> > > > **Response to Reviewer R1Eg**
> > > >
> > > > Thank you for the appreciation and increment of rating. As suggested, we plan to investigate the general nature of HYSP on other SSL methods and tasks in the future.
> > > > Regarding the batch size, we did not experiment on other methods. In fact, Tables 1, 2, and 4 report the performances from the original papers. However, we have found several notes on batch sizes in literature: SkeletonCLR [1], CrossCLR [1] and AimCLR [2] use a batch size of 128; P\&C [3], ISC [4] and MCC [5] use 64; 3s-Colorization [6], AS-CAL [7], and $MS^2L$ [8] use 32. So we have no reason to believe that they disregarded tuning this hyperparameter.
> > > >
> > > > ---
> > > >
> > > > [1] *3D Human Action Representation Learning via Cross-View Consistency Pursuit* (Li et al. CVPR 2021)
> > > > [2] *Contrastive Learning from Extremely Augmented Skeleton Sequences for Self-supervised Action Recognition* (Guo et al. AAAI 2022)
> > > > [3] *PREDICT & CLUSTER: Unsupervised Skeleton Based Action Recognition* (Su et al. CVPR 2020)
> > > > [4] *Skeleton-Contrastive 3D Action Representation Learning* (Thoker et al. ACM Multimedia 2021)
> > > > [5] *Self-supervised 3D Skeleton Action Representation Learning with Motion Consistency and Continuity* (Su et al. ICCV 2021)
> > > > [6] *Skeleton cloud colorization for unsupervised 3D action representation learning* (Yang et al. ICCV 2021)
> > > > [7] *Augmented skeleton based contrastive action learning with momentum LSTM for unsupervised action recognition* (Rao et al. Information Sciences 2021)
> > > > [8] *MS2L: Multi-Task Self-Supervised Learning for Skeleton Based Action Recognition* (Lilang et al. ACMMM 2020)

---

### Official Review · Reviewer_Dsnu · 2022-10-24

**Confidence:** 3
**Correctness:** 4
**Technical Novelty And Significance:** 3
**Empirical Novelty And Significance:** 3
**Recommendation:** 8

**Clarity, Quality, Novelty And Reproducibility:**

The paper is clearly written. I could follow it without prior knowledge of self-paced learning and hyperbolic embedding spaces. The technical contributions are sound and also make intuitive sense. The experimental results also highlight the novelty of the work. All parts of the method are fully described, making the work reproducible. The authors also provide their source code.

**Strength And Weaknesses:**

**Strengths**
1. The paper is clearly written and easy to follow. I was not familiar with prior work around hyperbolic embedding spaces and self-paced learning, but I found the concepts intuitive and the descriptions sound.

2. The experimental results clearly show the contributions of the individual components of the proposed method. The authors back up all their claims around curriculum scheduling, positive-only sampling, and uncertainty in self-paced learning with appropriate experiments.

3. I also found the limitations sections helpful to clearly understand the scope of the work.


**Weaknesses**
1. While I did not find any major concerns with the work, I am curious how the choice and the quality of positive samples affect performance. For example, how much does the performance depend on "hard positives"? Fig. 3a shows the correlation between the feature distances and the uncertainties. If only samples corresponding to the last few columns were to be used in the training (which I'm assuming are the hardest samples), how much would the performance drop? Since humans can perform similar actions with many diverse movements, real-world data can contain many such hard positives.

2. In addition to showing the inter-class variability (Sec. 4.4 and appendix), it can be illuminating to also see the confusion matrix, showing how the variability might be correlated with the per-class performance.

**Summary Of The Paper:**

The authors present a method to perform skeleton-based action recognition using a novel self-paced learning on features in a hyperbolic embedding space. The learning is "self-paced" in the sense that the magnitude of the gradients for the learning optimization directly depends on the certainty of the learned features. The authors follow a curriculum scheduling approach to stabilize their self-paced model. They also propose a positive-only learning paradigm, showing that adding negative samples into the hyperbolic embedding space deteriorates performance. Overall, the authors show appreciable improvements in action recognition performance using their hyperbolic self-paced model and also demonstrate the benefits of the individual components of their method through ablation studies.

**Summary Of The Review:**

The paper presents a sound and intuitive method for skeleton-based action recognition. The contributions are clear, and the experimental results sufficiently back up the authors' claims.

---

> ### Author Response · Authors · 2022-11-18
> **Response to Reviewer 1**
>
> 1. Thanks for your suggestion. We have experimented to compare the performance variation between using harder positives only (i.e. the samples with higher uncertainty) Vs. easier positives only (those samples with lower uncertainty).
> For the sake of comparison, we selected an equal number of hard and easy positives, namely the first and second half of the total samples, after sorting them in decreasing order of uncertainty (as estimated upon the HYSP pre-training on all samples). It turns out that learning with the harder examples yields the top-1 accuracy of 70.9\%, while the easier ones yield 74.5\%. It is expected that both results are well below the performance of the model trained on the full dataset (82.6\% with the joint stream), because of using half the amount of data.
> As for the performance discrepancy, we speculate that easier samples have an edge on the half-sized training set, while harder samples better support when larger amount of data is available. Let us also add that, as expected, the two new trainings have different durations: that one with harder samples converges and overfits earlier (at epochs 400 and 500 respectively) than with easier (epochs 400 and 700 correspondingly).
>
> 2. Thanks for the useful suggestion.
> We have computed the confusion matrix (see Figure 7 in Appendix A), arranging the actions by the median radius of the class embeddings. So, actions in the rows and columns are sorted according to their median radii (increasing left-to-right and top-to-bottom), using the values from Appendix A, Fig. 4.
> One observes that lower radii (bottom right corner) yield larger prediction errors among the actions. This is the case for actions such as writing (\#11) and brushing teeth (\#2), i.e. these are more ambiguous/uncertain because they are characterized by less distinct motion. (The action IDs are illustrated after the confusion matrix.) This study has been added to Appendix A.

---

> > ### Comment · Reviewer_Dsnu · 2022-11-22
> > **Reviewer Response**
> >
> > Thank you for the detailed response. I do not have further questions and maintain my original recommendation.

---

### Decision · Program_Chairs · 2023-01-20

**Decision:**

Accept: poster

**Justification For Why Not Higher Score:**

Two reviewers pointed out that parts of the approach already exist in prior works. Hyperbolic space and BYOL have been used for skeleton action recognition before.


**Justification For Why Not Lower Score:**

The overall method is interesting and novel. Also the results show the efficacy of such a design.

**Metareview: Summary, Strengths And Weaknesses:**

This work introduces a self-supervised representation learning method for skeleton action data, which is based on BYOL and the hyperbolic space. The proposed model achieves state-of-the-art performance on three datasets. The idea of self-pacing for self-supervised  skeleton representation learning is interesting, and the authors designed a novel approach based on hyperbolic geometry to tackle this. All reviewers recommend acceptance for this paper due to the novelty and efficacy of the method. Authors need to improve the camera ready version following reviewers' comments.

**Note From Pc:**

if the above contains the word "oral" or "spotlight" please see: "oral" presentation means -> notable-top-5% and "spotlight" means -> notable-top-25%. As stated in our emails, we are disassociating presentation type from AC recommendations